# Food Image Segmentation Using Multi-Modal Imaging Sensors with Color and Thermal Data

**DOI:** 10.3390/s23020560

**Published:** 2023-01-04

**Authors:** Viprav B. Raju, Masudul H. Imtiaz, Edward Sazonov

**Affiliations:** 1Department Electrical and Computer Engineering, The University of Alabama, Tuscaloosa, AL 35487, USA; 2Department Electrical and Computer Engineering, Clarkson University, Potsdam, NY 13699, USA

**Keywords:** dietary assessment, thermal imaging, image segmentation, food image segmentation, sensors, multi-modal sensors

## Abstract

Sensor-based food intake monitoring has become one of the fastest-growing fields in dietary assessment. Researchers are exploring imaging-sensor-based food detection, food recognition, and food portion size estimation. A major problem that is still being tackled in this field is the segmentation of regions of food when multiple food items are present, mainly when similar-looking foods (similar in color and/or texture) are present. Food image segmentation is a relatively under-explored area compared with other fields. This paper proposes a novel approach to food imaging consisting of two imaging sensors: color (Red–Green–Blue) and thermal. Furthermore, we propose a multi-modal four-Dimensional (RGB-T) image segmentation using a k-means clustering algorithm to segment regions of similar-looking food items in multiple combinations of hot, cold, and warm (at room temperature) foods. Six food combinations of two food items each were used to capture RGB and thermal image data. RGB and thermal data were superimposed to form a combined RGB-T image and three sets of data (RGB, thermal, and RGB-T) were tested. A bootstrapped optimization of within-cluster sum of squares (WSS) was employed to determine the optimal number of clusters for each case. The combined RGB-T data achieved better results compared with RGB and thermal data, used individually. The mean ± standard deviation (std. dev.) of the F1 score for RGB-T data was 0.87 ± 0.1 compared with 0.66 ± 0.13 and 0.64 ± 0.39, for RGB and Thermal data, respectively.

## 1. Introduction

Imaging sensor-based dietary assessment involves several stages of processing and algorithms such as food detection and identification [1,2], food segmentation, and food portion size estimation [3,4]. Two major segments of dietary assessment are food type recognition and food portion size estimation. Food segmentation is a necessary step when there are two or more food items as part of the meal. This involves identifying and isolating food items in meal images which is known as the segmentation problem. The two main categories of segmentation are semantic segmentation and instance segmentation. Semantic segmentation is implemented by associating every pixel of the image to a class. Here, multiple objects of the same class are treated as a single entity. Contrastingly, instance segmentation treats multiple objects of the same class as distinct individual instances. This paper aims to separate food items of the same type (semantic segmentation).

Food image segmentation can either be performed manually or by automatic computer vision methods. In contrast to manual food recognition, manual segmentation is very time-consuming due to the need to draw accurate food contours on the images by hand. Automatic food segmentation methods have become more popular, replacing manual methods. Once food objects are segmented, each food object’s volume (or portion size) can also be estimated. In this paper, we focus on the problem of food segmentation. We aimed to tackle segmentation cases such as:Two or more foods placed in the same container.Foods that are of the same texture and/or color.Food combinations that mimic real-world scenarios.

Typically, food images are color images with three color channels (RGB). When there is an occurrence of similar-looking foods, for example, leafy steamed vegetables and salads, it is challenging to distinguish between the two because of their similar texture and color attributes. 

Food image segmentation is relatively under-explored and there have been very few related-studies in this field. In [5], a threshold-based approach that converts a color food image into a high-contrast grayscale image was presented. The dataset in this study consisted of single food items on a flat background. The authors in [6] explore methods to improve segmentation results on food images. The method is based on contour initialization through classical snakes (active contours). Another paper discusses food segmentation from images acquired by a wearable camera. A saliency-aware Active Contour Model (ACM) for automatic segmentation was utilized [7]. They achieved accuracies higher than conventional segmentation methods. However, multiple food items on the same plate were not considered. Another study involved Local Variation (a graph-based approach) in generating a list of initial segmentations [8]. A semi-automatic approach was discussed in [9], where a user was expected to draw bounding boxes on the food image, using which the system segmented each food item region by Grub-Cut. A study [10] utilized a deep CNN-based food border detection in sequence with region growing methods. The paper also talks about a semi- automatic method that requires minimal user input. The two dish segmentation methods achieved average segmentation accuracies of 88% (automatic) and 92% (semi-automatic), respectively, on a dataset of 821 images. The authors in [11] proposed a segmentation method based on normalized cut and super-pixels. The method relied on color and texture cues. The same research group proposed a graph-based segmentation method with results similar to the previous study [12]. The authors in [13] discuss an automatic segmentation method for food images using a deep convolutional neural network. The training set consisted of 1027 tray images while the test set consisted of 390 plate images. The pixelwise fully automatic segmentation method attained 91.2% intersection over union (IOU) compared with the semi-automatic graph-cut method with an IOU of 93.7%. 

Researchers have also explored Hyper-spectral imaging as an alternative to true-color-based dietary assessment [14]. The authors evaluated the fusion of RGB and hyperspectral features and found that the fused feature set was significantly more predictive of within-food caloric content compared with the individual RGB and hyperspectral feature sets.

K-means image segmentation has been used previously to segment food images. One such application was in fruit processing that involved infected fruit part detection [15,16]. Banana images were segmented using a two-stage k-means clustering methodology [17]. The two stages segmented contours of the whole banana and the damaged lesions/senescent spots on the banana surface, respectively. The authors in [18] proposed an adaptive k-means image segmentation technique to identify an optimal value of K based on the number of connected domains. Another study proposed the use of a Sobel operator for automatic food product segmentation. Sobel operator was used prior to k-means clustering to determine a region of interest (ROI) following which the object and background in the ROI are separated [19]. 

The studies described above did not consider instances where similar-looking foods were present on the same plate. The proposed methodology attempts to address this issue by utilizing multi-modal thermal and color imaging sensors. 

Multi-modal imaging sensors, specifically color, infrared and thermal imaging, have been popular among researchers. One study explored multi-source data fusion to monitor wheat powdery mildew [20]. Texture features were extracted from the thermal infrared images and RGB images of wheat with powdery mildew and used as input to build a prediction model for a wheat powdery mildew disease index. Another paper [21] presents a method to segment human hands and objects held in those hands using a multimodal 3D data (3D point cloud, RGB, thermal) to implement a safe human–robot handover. The study also presents a solution to tackle the calibration and alignment issues while using multiple cameras. A set of cameras, namely, RGB, thermal, and NIR cameras were used and a copper–plastic chessboard calibration target with an internal active light source (near-infrared and visible light) was used for calibration. The multi-modal dataset showed a significant improvement over other sets. Thermal imaging was used to detect downy mildew on grapevine leaves [22] and to study the occurrence of diabetic foot ulceration by segmenting plantar foot thermal images [23]. Another study [24] fused thermal sensor data with radar data for vehicle tracking. The thermal sensor proved to be versatile under different weather conditions. One study used the fusion of images from a stereo camera and the thermal images from a thermal sensor for building inspection [25]. A novel approach to geometrically calibrate and pixel-co-register the trifocal camera system was presented here by the authors. 

Some of the studies that involve multi-modal data have used neural networks to process the data for various applications. Since neural networks require a considerable amount of data for training, we looked towards classical methods for our application. In this study, k-means clustering was adopted. However, two major drawbacks of these clustering-based methods, as noted by the authors in [26], are: (1)Lack of variability in image attributes, such as color, intensity, and texture, might reduce the performance of these methods.(2)The number of clusters is not known; such an unsupervised clustering scheme may not produce an optimal number of segments.

To solve the two drawbacks of clustering methods, we propose two solutions. We propose a new approach that uses an additional channel of information—temperature. This approach introduces a fourth dimension to the food image. The segmentation now would not only look at visual features of the food but also at its temperature. Adding the temperature information would increase the variability of the data. To find the number of clusters in the image we utilize a cluster size optimization technique (Within-cluster Sum of Squares). The contributions of our paper are as follows: (1) a novel RGB plus thermal food-imaging system and (2) a novel four-dimensional food-image segmentation.

The remainder of this paper is organized as follows: Section 2 presents a description of the equipment, data, and the methods used. Section 3 and Section 4 are the results and discussion, respectively, followed by the conclusion in Section 5.

## 2. Methods

### 2.1. Equipment

The study involved a custom-made imaging system based on a Raspberry Pi Model 3B running Raspbian Stretch 4.14 operating system with a Pi RGB Camera (V2) and a FLIR Lepton 3 thermal camera (Figure 1). A 5-megapixel OV5647 sensor was used for RGB imaging. The image resolution used for image capture was 1024 x 768 (aspect ratio = 4:3). The FLIR Lepton 3 is a long-wave infrared (LWIR) camera module. The camera is interfaced using an SPI (plus a 2-wire I2C control) interface and captures infrared radiation input in the wavelength band between 8 and 14 microns. The thermal image could be stored in two formats—a text file (.txt) with raw temperature readings or an RGB image (.png format) that consists of raw data processed by the interface to be represented as a color image. The raw file was used in this study and will be referred to as ‘thermal image’ hereafter. The resolution of the thermal image was 160 x 120 (aspect ratio= 4:3). Both cameras and the Pi were mounted on a wooden plank and fixed to a camera support system.

### 2.2. Data

The integrated Red-Green-Blue-Thermal (RGB-T) imaging system was used to capture images of various food items served on a paper plate. The imaging system was placed at approximately 35 cm from the plate.

Food combinations used in the study matched real-world scenarios that contained a mixture of hot foods, cold foods, and foods at room temperature. There were mainly six combinations of food items (Table 1). For each combination, one RGB and one thermal image were captured. The data were divided into a temporal dataset (Combination C1) and an atemporal dataset (Combinations C2-C6). In Combination C1, a temporal sequence of data was captured every 15 s in the period t ε (0 s to 150 s) to demonstrate the cooling process and quantify the changes in performance. The set consisted of 11 pairs (thermal and RGB) of images. 

The atemporal dataset consisted of 5 pairs of images, one pair for each combination. All images were captured under fluorescent lighting (lux reading between 350 and 400 lx) conditions in an indoor kitchen. 

### 2.3. Camera Calibration

Camera calibration is an essential step of monocular and multi-view computer vision applications. Calibrating cameras will ensure uniformity across all imaging sources used in the experiment. The main purpose of calibration is to eliminate any distortions in the captured image. Distortion is introduced to the cameras due to the manufacturing process. Barrel distortion is an aberration where straight lines are warped inwards in the shape of a barrel. The phenomenon occurs when field of view of a lens is much wider than the size of the imaging sensor. Calibrating using a known reference pattern could reduce such distortions [27,28]. In [27], the authors used a checkerboard pattern to calibrate a RGB camera. RGB sensors can sense simple patterns printed on paper, provided that there is sufficient lighting. Thermal calibration, however, requires certain elements that the sensor can sense. These elements should have temperatures that are distinct from their surroundings. Previously, researchers have utilized calibration grids made of copper lines on a PCB [29], wires [30], bulbs [31,32] and metal plates [30]. In [30], a light-heating method using a checkerboard pattern was proposed. This method utilizes the absorption property of the black boxes of the checkerboard.

The black boxes absorb more heat and would be distinctly sensed by the thermal sensors compared with the white boxes. The same calibration method was used in this study. A checkerboard image printed on paper was placed under sunlight and used as a common calibration pattern for RGB and thermal cameras. The camera parameters were stored and used to undistort images (Figure 2) captured during the study. Image rectification was performed once both the cameras were calibrated, as in [28]. The rectification process is depicted in Figure 3.

### 2.4. Image Registration

To obtain the combined RGB-T image from the two individual RGB and thermal images, the two images need to be accurately superimposed on each other. This superimposition of multi-sensor images can be achieved using image registration. Registration of multi-sensor images is a very important process in many computer vision applications such as remote sensing [33] and medical image analysis [34,35]. Control-point-based (CP) [36,37,38] and intensity-based [39,40] algorithms are two popular image registration methods. Once registration is completed, one of the images is usually transformed and projected on the other (superimposed).

In this study, the CP-based method was chosen with a projective transformation. Since the two cameras were mounted on a fixed setup, registration was performed only once, and the transformation object was stored. The initial registration was carried out using two checkerboard images (one from the RGB camera and one from the thermal camera). A sunlight-heated checkerboard was used for this purpose, and the control points were manually picked (using the corners of the checkerboard). The stored transformation object was used for all pairs of RGB and thermal images in the study. The thermal images were projected onto the RGB images to obtain a four-dimensional RGB-T image.

### 2.5. K-Means Clustering-Based Image Segmentation

For the k-means problem [41], the known variables are an integer k and a set of n data points X ⊂ ℝ. The goal is to choose k centers -C to minimize the potential function,
(1)$$\varphi \;=\underset{x\in X}{\sum }\underset{c\in C}{\min}\|x-c{\|}^{2}$$

By choosing the centers, one can implicitly define a set of clusters, one for each center. One cluster is a set of data points that are closer to a center than to any other center. The k-means clustering algorithm is as follows [42]:Randomly choose k initial centers C = {c1, …, ck}.For each i ∈ {1, …, k}, set the cluster ci to be the set of points in X that are closer to ci than they are to cj for all j ≠ i.For each i ∈ {1, …, k}, set ci to be the center of mass of all points in ci.Repeat Steps 2 and 3 until C no longer changes.

One limitation of the k-means clustering algorithm is that the number of clusters must be selected at the start. There are several optimization techniques that determine the optimal number of clusters. One such method uses the Within-cluster Sum of Squares (WSS) [43,44,45]. WSS is the sum of the squared deviations from each observation and the cluster centroid. A higher WSS represents a greater variability of the observations within the cluster. The total within-cluster sum of square (WSS) represents the compactness of the clustering, and we want it to be as small as possible. After a certain point, the WSS tends to saturate, i.e., adding more clusters will not improve the total WSS. An algorithmic way to determine this saturation point is the elbow method [46]. We used the elbow method to identify the best number of clusters for the k-means algorithm. K-means algorithm was applied for each set of images (RGB, thermal and RGB-T) in all six combinations. For each image, the k value was varied, in the range k ∈ (2: 10). WSS as a function of k was plotted for each image, and the optimal clusters were identified using the following algorithm:Run the clustering algorithm (e.g., k-means clustering) for different values of k. Here, we vary k from 2 to 10 clusters.For each value of k, calculate the total within-cluster sum of square (WSS).Plot the curve of WSS according to the number of clusters k.The bend (knee) location in the plot is generally considered an indicator of the appropriate number of clusters (for all cases in the thermal data). In case of an ambiguous bend, where there was no distinct bend in the plot, a tangent was drawn on the curve starting from the last point (k = 10). The point of tangency with the lowest k value is chosen automatically. This value was selected as plots tend to saturate (in terms of WSS) at this point.

Appendix B consists of the WSS plots for each of the three images (RGB, thermal, and RGB-T) of the six combinations. The number of optimal clusters selected for each case are also presented. The input used for K-means are the intensity values for R-G-B and the thermal sensor reading T from the thermal camera for each pixel in both cases. Additionally, for the combined case, the thermal value for each pixel is combined with that pixel’s RGB intensity value. After segmentation, the segmented image may consist of some unwanted regions, discontinuities, or noise. Post-processing of the segmented image to remove noise is often recommended by researchers. Here, median filtering was used for removing noise in the k-means segmented image. Once the segmented regions are obtained, a manual reviewer identifies and selects the food regions.

### 2.6. Segmentation Evaluation

For evaluation, manually annotated segmentation labels were used as the ground truth. The measure of evaluation used here is the F1 score (also known as Dice Similarity Coefficient). Segmentation of every occurrence of a food item in the dataset was evaluated.

The F1 score for two sets, *A* and *B*, can be calculated as:(2)$$F1\;\mathrm{score}=2\times \frac{\left|intersection\;\left(A,B\right)\right|}{\left(\left|A\right|+\left|B\right|\right)}$$
where *A* and *B* represent the ground truth and predicted segmentation for a food item.

## 3. Results

The segmentation results for each occurrence of a food item are summarized in Table 2 and Table 3. The final segmentation masks for three types of data (RGB, thermal, and RGB-T) are depicted in Figure 4. The F1 scores (mean ± std. dev.) for RGB-T, RGB, and thermal data were 0.87 ± 0.11, 0.66 ± 0.13 and 0.64 ± 0.39, respectively.

## 4. Discussion

This paper evaluated a novel food-imaging sensor system and an associated food image segmentation approach. The method combines RGB data and thermal data to obtain a four-dimensional RGB-T image. The purpose of this approach was to tackle segmentation issues that persisted in other methods of food image segmentation. One issue was multiple food items placed on the same plate. This study tested out six food combinations, two food items in each combination, served on the same plate.

The first combination consisted of multiple sets of images taken at 15-sec intervals. This combination was used to test the temporal aspect of the RGB-T data. The images had the same food items, but the hot item that was served earlier (French fries) cooled down to room temperature. The accuracy of the thermal segmentation deteriorated over time as the fries cooled down. Figure 5 depicts the plot of F1 score vs. time for RGB-T, RGB, and Thermal data in the case of French fries.

In C2, the RGB-T data produce consistent results compared with the other two types of data. In C2, steamed vegetables consisting of carrots and potatoes (served from a generic brand of steamed potato and carrots in a bag) and frozen fruits were used. The RGB data perform poorly in this case. However, carrots and potatoes appear to be distinct visually, but the ground truth labels for the vegetables considered them as one food item because they were served from the same bag. The segmentation results were not expected to be good for this combination. RGB-T data use thermal information and segments the regions similar to the ground truth label. This suggests that looking at the thermal information might provide insight into the type of food, along with the temperature and visual information.

In C3 and C4, RGB data result in very poor segmentation results due to the high similarity between the two food items. The thermal data offer no distinctive features for the potato chips or the salad, as they are both at room temperature and blend into the background. In the case of RGB-T data, the thermal features of the French fries add some variability to the original RGB information. This enables the algorithm to distinguish between the two food items in both combinations.

In C5 and C6, both RGB-T and thermal data produce good, acceptable results while the RGB data fail again with poor segmentation results.

As a final step, we processed the RGB images using some of the popular image segmentation methods [47,48,49,50,51,52] and the results are depicted in Figure 6. These color-based methods are limited to the three-dimensional RGB data and cannot be used without modification to segment the multi-modal RGB-T data. Therefore, we compare these results with the RGB-T K-means segmentation. RSFCC [51] presented the best results but failed for combinations C3 and C5.

The overall segmentation results help us make the following observations:RGB data may perform poorly when the foods have the same color, texture, and/ or intensities.Thermal data may perform poorly when foods are at room temperature as the food blends into the background.Thermal data perform better when hot and cold combinations are present.

The contributions of our paper are as follows: (1) a novel RGB plus thermal food-imaging system and (2) a novel four-dimensional food-image segmentation. Some of the previous methods that involved food image segmentation did not consider similar-looking foods that were present on the same plate. The proposed methodology attempts to address this issue by utilizing thermal imaging. To the best of our knowledge, thermal sensors have never been used before for diet monitoring. Therefore, assessing the thermal features of the food items is a novelty in itself. Future-work could include exploring the thermal properties of food items and study how they affect the behavioral metrics of a person’s dietary intake.

One limitation of this paper is the number of test cases used. We intend to integrate a thermal camera with a RGB in a single enclosure that can be packaged and carried around to monitor food intake with various functionalities, such as portion size estimation, food recognition, segmentation, etc. This article only discusses the segmentation part as a pilot study, and therefore is limited to the provided examples. Therefore, future studies could include a well-packaged device that can collect data on the go as users can track and monitor their intake and potentially, a larger dataset.

Another limitation of the proposed method is that the final segmented food region is selected by a trained assistant which can be automated in future studies. Additionally, a very limited number of food items were tested. A more complex dataset with more variety in the type of foods and number of foods could be tested.

A major point to be noted is that there are very few classical methods that could segment four-dimensional data (RGB-T in this case). Additionally, therefore, we used the simplest method to test our pilot data. Future research could involve more complex segmentation algorithms. The more recent neural networks have been reported to outperform the classical methods. Given sufficient data, a neural network can be trained to perform the segmentation task. Using K-means clustering for image segmentation would not be the present-day standard such as deep-learning-based methods [53]. However, since our system is unique and there are limited data available, the neural network approach is not feasible. The lack of sufficient samples led us to take the conventional way. A future direction could be to collect more RGB-T data and test neural networks for segmentation.

## 5. Conclusions

In this paper, a novel approach to food imaging was proposed. The acquisition and processing of four-dimensional (RGB-T) data were presented. We tried to segment several combinations of foods at different temperatures (above, below, and at- room temperature) that mimic real-world scenarios. Adding the temperature information increased the variability of the data leading to better segmentation results. A cluster size optimization technique was utilized to identify the number of optimal clusters in each image. The contributions of our paper are as follows: (1) a novel RGB plus thermal food-imaging system and (2) a novel four-dimensional food segmentation. Using k-means clustering-based image segmentation, the RGB-T data produced better results in terms of F1 score when compared with the original RGB and thermal data. The RGB-T image segmentation achieved an F1 score of 87 ± 0.11.

## Figures and Tables

**Figure 1 sensors-23-00560-f001:**
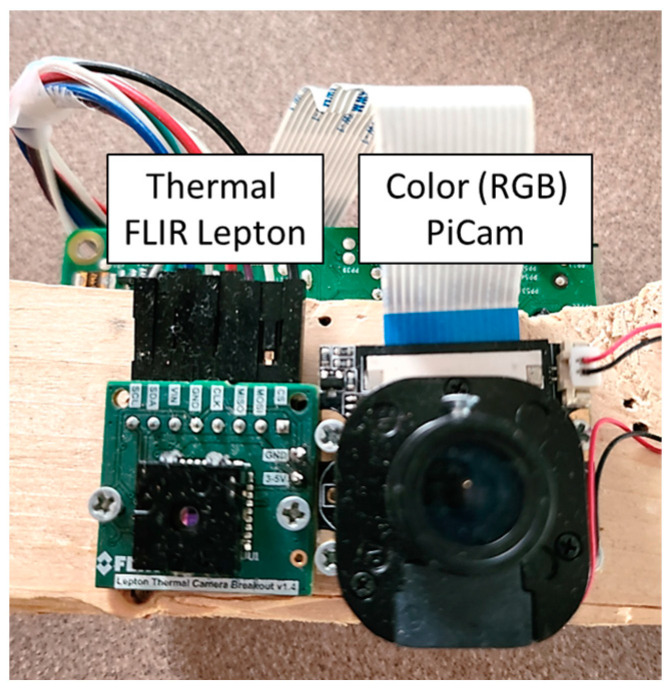
The RGB-T Raspberry Pi imaging system that was used for data acquisition.

**Figure 2 sensors-23-00560-f002:**
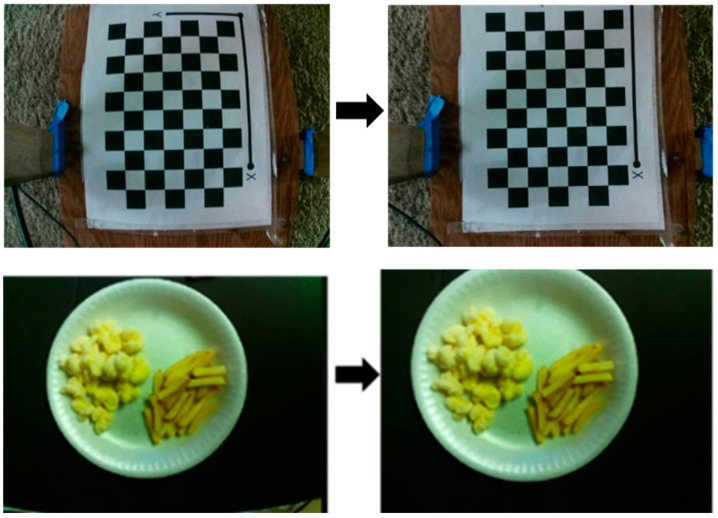
Camera lens distortion correction. The distortion coefficients are calculated using the checkerboard images. The figure indicates before and after images of the correction model.

**Figure 3 sensors-23-00560-f003:**
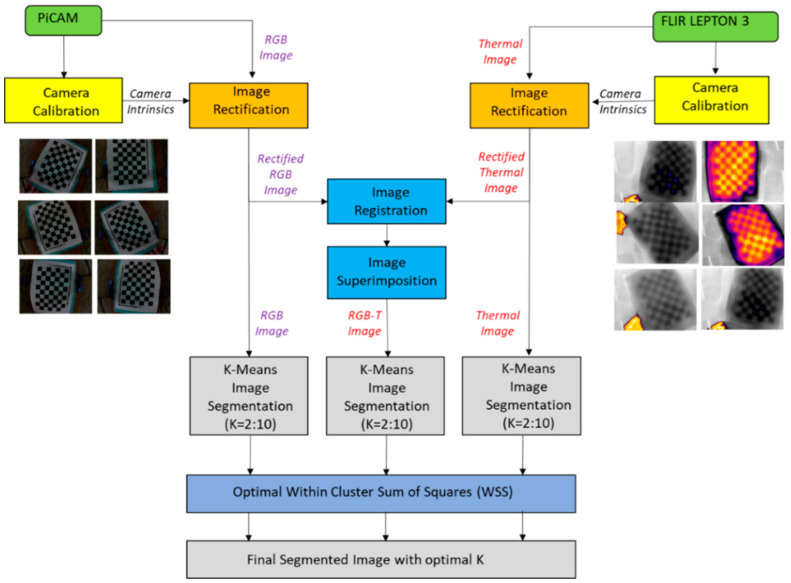
Proposed system framework for acquisition and processing of RGB, thermal and RGB-T images.

**Figure 4 sensors-23-00560-f004:**
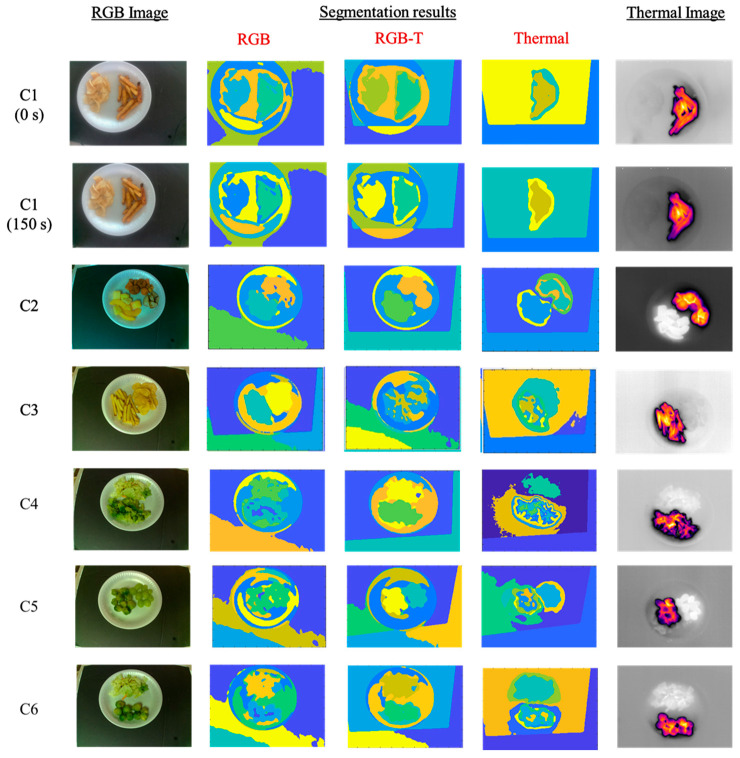
Segmentation results of the six combinations using RGB, thermal and RGB-T data.

**Figure 5 sensors-23-00560-f005:**
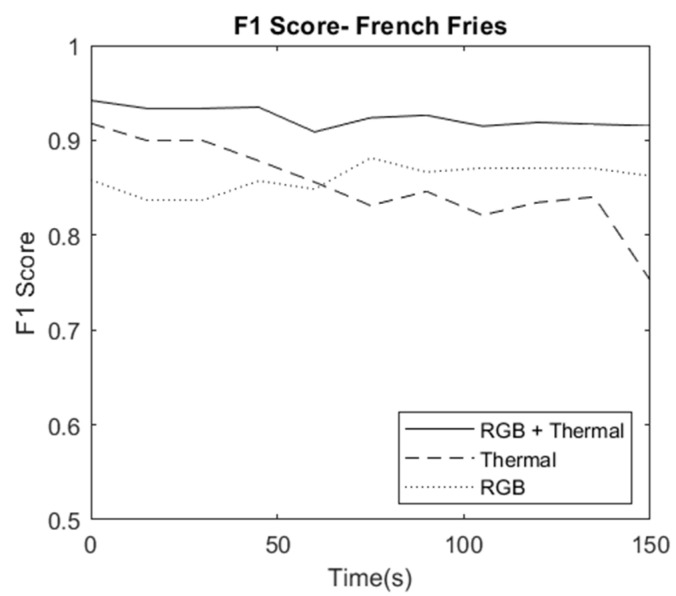
Segmentation Evaluation for the temporal dataset: F1 Score vs. Time (French Fries in Combination C1).

**Figure 6 sensors-23-00560-f006:**
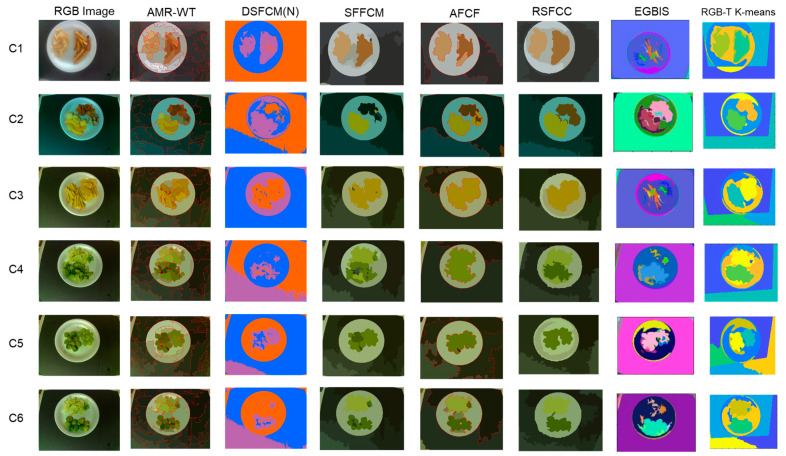
Segmentation results of the six combinations of RGB images, using some of the popular color segmentation methods: AMR-WT [47], DSFCM(N) [48], SFFCM [49], AFCF [50], RSFCC [51], EGBIS [52], compared with the results obtained using the RGB-T K-means segmentation.

**Table 1 sensors-23-00560-t001:** Dataset: 6 Combinations of Food Items.

Combination	Hot Food (>30 °C)	Warm Food (29–15 °C)	Cold Food (<15 °C)
C1 (Initial—0s)	French Fries	Potato Chips	-
C1 (Final—150 s)	-	Fr. fries & Potato Chips	-
C2	Steamed Veggies	-	Fruits (Tropical)
C3	French Fries	Corn Chips	-
C4	Steamed Broccoli	Salad	-
C5	Steamed Brussel Sprouts	-	Fruits (Grapes)
C6	Steamed Brussel Sprouts	-	Salad

**Table 2 sensors-23-00560-t002:** Segmentation Results (Combinations C1–C6): F1 Score.

Combination	Food Item	F1 Score (Dice Coefficient)		
		RGB-T	RGB	Thermal
C1 (Mean)	Fries	**0.859**	**0.754**	**0.743**
	Potato Chips	**0.549**	**0.544**	~
C2	Veggies	**0.922**	**0.785**	**0.886**
	Fruits	**0.936**	**0.945**	**0.879**
C3	French Fries	**0.924**	**0.479**	**0.882**
	Corn Chips	**0.92**	**0.531**	~
C4	Broccoli	**0.925**	**0.659**	**0.886**
	Salad	**0.766**	**0.659**	~
C5	Brussel Sprouts	**0.904**	**0.551**	**0.857**
	Grapes	**0.893**	**0.619**	**0.809**
C6	Brussel Sprouts	**0.934**	**0.732**	**0.928**
	Salad	**0.864**	**0.693**	**0.845**
IOU (Mean ± Std. Dev.)		0.87 ± 0.11	0.66 ± 0.13	0.64 ± 0.39

Color code (Ranked for each row): Green = Highest, Blue = 2nd Highest, Red = Lowest, ~ = No meaningful information.

**Table 3 sensors-23-00560-t003:** Segmentation Results (Temporal Dataset): F1 Score.

Time (s)	Food Item	Thermal + RGB	RGB	Thermal
0	French Fries	**0.89**	**0.752**	**0.848**
	Potato Chips	**0.552**	**0.553**	~
15	French Fries	**0.867**	**0.717**	**0.791**
	Potato Chips	**0.554**	**0.557**	~
30	French Fries	**0.875**	**0.719**	**0.818**
	Potato Chips	**0.543**	**0.522**	~
45	French Fries	**0.878**	**0.75**	**0.783**
	Potato Chips	**0.526**	**0.56**	~
60	French Fries	**0.833**	**0.737**	**0.748**
	Potato Chips	**0.55**	**0.542**	~
75	French Fries	**0.858**	**0.788**	**0.711**
	Potato Chips	**0.555**	**0.522**	~
90	French Fries	**0.863**	**0.764**	**0.733**
	Potato Chips	**0.567**	**0.563**	~
105	French Fries	**0.843**	**0.771**	**0.697**
	Potato Chips	**0.568**	**0.567**	~
120	French Fries	**0.85**	**0.771**	**0.716**
	Potato Chips	**0.529**	**0.53**	~
135	French Fries	**0.847**	**0.77**	**0.724**
	Potato Chips	**0.54**	**0.517**	~
150	French Fries	**0.844**	**0.758**	**0.603**
	Potato Chips	**0.552**	**0.546**	~

Color code (Ranked for each row): Green= Highest, Blue= 2nd Highest, Red= Lowest, ~ = No meaningful information.

## Data Availability

Data collected by the RGB-T assembly is provided with this article as Appendix A. RGB images and thermal sensor images used in this study are provided. This includes any calibration data that was used.

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
