# Peer review of "Food Image Segmentation Using Multi-Modal Imaging Sensors with Color and Thermal Data"

_sensors, 2023, doi:10.3390/s23020560_

Round 1

Reviewer 1 Report

The paper is well written and the topic for combining multi-modal data to the food analysis is new.

While at the current stage, I think the device cost need to be considered for its real applicability. But it is fine for a research paper.

I suggest the acceptance of this paper as is.

Author Response

We thank the reviewer for their comments. Cost is definitely a valid point. We intend to integrate a thermal camera and an RGB in a single enclosure that can be packaged and carried around to monitor food intake with various functionalities, such as portion size estimation, food recognition, segmentation, etc. This article only discusses the segmentation part as a pilot study and is limited to the provided examples. Therefore, future work could include a well-packaged device that can collect data on the go as users can track and monitor their intake. 

Reviewer 2 Report

The paper aims to address the challenging multi-food segmentation problem by using RGB and Thermal information. The work is very interesting and below are some comments that can help improve the quality:

(1) The introduction should also include some existing methods that for general image data instead of food, are those method work or not work for food data and what is the potential challenge. In addition, the author should better describe the objective of this work by introducing instance segmentation and semantic segmentation, which are the two main categories of segmentation methods. 

(2) Though Section 2.5 describes the K-means algorithm, is not clear how it is related to the food region segmentation. E.g. What are the input used for clustering? pixel or feature vector? Those needs more detail description. 

(3) I understand the dataset preparation can be difficult, but only 6 food combinations with total 10 food types make the experimental part less convincing. If possible, I suggest to include more foods. 

(4) Currently the experimental results can only show RGB-T works better than RGB and Thermal. Therefore, it would be better if the author can include some existing segmentation methods for comparison, the existing method may not need to be food focused or K-mean based methods. 

(5) The quality of figure can be improved. E.g. the legend in figure 2 is almost invisible. In addition, there are some typos in this paper, the author needs to read again and correct them. Also, in table 2, one of the "~" is in blue color, I believe this is also a mistake. 

Reviewer 3 Report

The purpose of this study is to present a novel approach to food imaging which comprises of two imaging sensors, which are color-based (Red-Green-Blue) as well as thermal-based. This paper has a very interesting topic and its objectives and contents fit the journal well. I have two specific comments.

1- It would be useful to discuss the practical applications of this method.

2- In the discussion section, it is necessary to highlight the innovations and advancements made in the methodology compared to previous studies.

Reviewer 4 Report

Interesting subject. Homemade RGB and thermal camera combinations can be used for student experimentation. 

The paper, as it is, cannot be accepted. Even if the hardware is interesting and the objective, which is the segmentation of food images, is of great interest, the K-mean method does not give satisfactory results. The idea of using an RGB and an IR image is of interest.
Nowadays, there are many segmentation methods. One idea is using the information about the number of the different food present. Spatial correlation information as well as the texture can also be used. 

Many other ML segmentation methods are nowadays available. A more extensive comparison can be done on the authors data base. 

So, overall, the paper must be improved greatly before its acceptance. In particular, many repeated sentences have to be removed. 

Also, if the authors can share their database, this can be a positive point for their paper. 
